Subject Areas:
ecology

Keywords:
arthropods, diversity, fertilization, litter, *Nasutitermes*, *Trigona spinipes*

Author for correspondence:
Germano Leão Demolin Leite
e-mail: gldleite@ica.ufmg.br

# Diversity of arthropods on *Acacia mangium* (Fabaceae) and production of this plant with dehydrated sewage sludge in degraded area

Júlia Leticia Silva[1], Germano Leão Demolin Leite[1],
Wagner de Souza Tavares[2], Farley William Souza Silva[3],
Regynaldo Arruda Sampaio[1], Alcinei Mistico Azevedo[1],
José Eduardo Serrão[4] and José Cola Zanuncio[5]

[1]Instituto de Ciências Agrárias, Universidade Federal de Minas Gerais, 39404-006 Montes Claros, Minas Gerais, Brasil
[2]Asia Pacific Resources International Ltd. (APRIL), PT. Riau Andalan Pulp and Paper (RAPP), Pangkalan Kerinci, Riau, 28300 Sumatra, Indonesia
[3]Centro de Ciências Biológicas e da Natureza, Universidade Federal do Acre, 69920-900 Rio Branco, Acre, Brasil
[4]Departamento de Biologia Geral and [5]Departamento de Entomologia/BIOAGRO, Universidade Federal de Viçosa, 36570-900, Viçosa, Minas Gerais, Brasil

JLS, 0000-0002-4394-1578; GLDL, 0000-0002-2928-3193;
WdST, 0000-0002-8394-6808; FWSS, 0000-0002-2651-9375;
RAS, 0000-0003-3214-6111; AMA, 0000-0001-5196-0851;
JES, 0000-0002-0477-4252; JCZ, 0000-0003-2026-281X

Sewage sludge is an organic matter-rich material with abundant fractions of nitrogen and other macro and micronutrients, essential for plant growth and development such as *Acacia mangium* Willd. (Fabales: Fabaceae) used in recovering actions of degraded areas. The objective of this study was to evaluate over 24 months the abundance and diversity of chewing and pollinator insects and arthropod predators on *A. mangium* plants and the mass production and soil coverage by this plant, fertilized with dehydrated sewage sludge, in a degraded area. The experimental design was in randomized blocks with two treatments (with and without dehydrated sewage sludge) and 24 replications. The number of leaves per branch and branches per plant, defoliation percentage by chewing insects, soil cover and abundance of chewing and pollinator insects and arthropod predators were higher on *A. mangium* plants fertilized with dehydrated sewage sludge. *Nasutitermes* sp. (Blattodea: Termitidae) and *Trigona spinipes* F. (Hymenoptera:

Apidae) were the most observed insects on trunks and leaves, respectively, of *A. mangium* plants fertilized with dehydrated sewage sludge. The *A. mangium* fertilization increases the populations of different insect and spider groups on this plant.

# 1. Introduction

Sewage sludge is the residual and semi-solid material produced as a by-product during sewage treatment of industrial or municipal waste [1]. It is rich in organic matter and nutrients with potential for reuse as a fertilizer and media for seedling production after processes of its stabilization such as anaerobic digestion and composting [2]. Crops cultivated using stabilized sewage sludge include the Japanese mustard spinach, *Brassica rapa* L. var. *perviridis* (Brassicales: Brassicaceae) in Japan [3]. Sewage sludge can be re-used as fertilizer in forest plantations, degraded area under recovery process and in agriculture, reducing production costs and environmental risks [4–6]. The quality of dried and pasteurized sewage sludge is classified as class A by the USA [7]. A treated sewage sludge from the 'Estação de Tratamento de Esgoto (ETE)' in the municipality of Juramento, Minas Gerais State, Brazil had no helminth eggs and protozoan cysts, and did not increase the heavy metal contents in grains of maize, *Zea mays* L. (Poales: Poaceae), and cowpea, *Vigna unguiculata* (L.) Walp. (Fabales: Fabaceae) [8].

Mangium, *Acacia mangium* Willd. (Fabales: Fabaceae), is native to northeastern Queensland in Australia, the Western Province of Papua New Guinea, Papua and the eastern Maluku Islands [9]. It is a fast growing, hardy and pioneering plant with nitrifying potential, which makes it suitable for degraded area recovering [10]. This plant is used to restoring wastelands created by open-pit gold mining in Colombia [11]. The high fix atmospheric nitrogen gas fixation by this plant in symbiosis with diazotrophic archaea and bacteria increases biomass productivity and nutrient inflow via litter, favouring ecological succession [12]. These characteristics and the high *A. mangium* adaptability to acidic, infertile and flooding-prone soils increase its potential to recover degraded areas [13,14].

Insect (Insecta) diversity, with known function, population or status of these organisms, is an indicator (i.e. bioindicator) that can reveal the qualitative status of the degraded area recovery, responding rapidly to environmental changes [15]. Coleoptera, Lepidoptera and Hymenoptera, with a large number of families and described species (around 400 000; 180 000 and 150 000 respectively), are widely used as bioindicators around the world [16–18]. Plant chemical composition and development (=age) affect the diversity of phytophagous insects and their natural enemies (insects and spiders (Araneae)) and, therefore, serve as nutritional and chemical defence indexes for plants [19,20]. Sewage sludge applied as a boosting material in crops increases the soil organic matter content, besides being rich in macronutrients such as calcium, magnesium, nitrogen and phosphorus, and micronutrients such as copper and zinc [21], favouring plant growth and development and interaction with insect ecology processes.

Hypotheses tested interactions among plant, dehydrated sewage sludge, degraded area and arthropod predatory organisms: plants fertilized with dehydrated sewage sludge have larger crown and litter formation [22,23]. More complex host individuals—larger trees—(i.e. biogeographic island theory (BGI)) support a higher pest insect abundance and diversity owing to better food availability [24,25] and, consequently, more arthropod natural enemies [26]. The tree canopy is a small-scale BGI and an example to test these hypotheses [27]. BGI predicts that extinction rates are higher in smaller islands because they cannot stand high organism populations with the rarest species being more vulnerable to extinction [28]. BGI considers the history of the biological processes such as colonization, speciation and extinction to explain species distribution patterns [29]. Smaller trees are likely to support small populations.

The objective of this study was to evaluate plant biomass production, soil cover by plants and litter and diversity and abundance of chewing and pollinator insects and arthropod predators on *A. mangium* plants, over 24 months, fertilized with dehydrated sewage sludge in a degraded area. The hypotheses tested were that *A. mangium* plants fertilized with dehydrated sewage sludge have larger canopy forming more litter, helping in the degraded area recovery (i); fertilized plants were bigger (>BGI) and had greater abundance of chewing (ii) and pollinator (iii) insects, and arthropod predators were more numerous on larger plants (iv).

# 2. Material and methods

## 2.1. Experimental site

The study was carried out in a degraded area at the 'Instituto de Ciências Agrárias (ICA)' of the 'Universidade Federal de Minas Gerais (UFMG)', municipality of Montes Claros, Minas Gerais State, Brazil (latitude 16°51′ S × longitude 44°55′ W, altitude 943 m) from March 2015 to February 2017 (24 months; arthropod collection period). The area was defined as degraded owing to soil losses and changes in soil chemistry or hydrology [30,31]. The climate of this region is Aw: tropical savannah, with dry winter and rainy summer, according to the Köppen classification [32]. The soil type is litolic neosoil [33] with average texture, total sand = 42.0 dag $Kg^{-1}$, silt = 36.0 dag $Kg^{-1}$, clay = 22.0 dag $Kg^{-1}$, pH–$H_2O$ = 5.0, organic matter = 4.4 dag $Kg^{-1}$, P = 1.5 mg $dm^{-3}$, K = 92.0 mg $dm^{-3}$, Ca = 1.9 $cmol_c$ $dm^{-3}$, Mg = 0.8 $cmol_c$ $dm^{-3}$, Al = 2.4 $cmol_c$ $dm^{-3}$, H + Al = 6.7 $cmol_c$ $dm^{-3}$, cation–exchange capacity (CEC) = 5.3 $cmol_c$ $dm^{-3}$ and CEC at natural pH 7.0 = 9.6 $cmol_c$ $dm^{-3}$ after soil chemical and physical analysis carried out in 2014 in a laboratory using standard international protocols [34].

## 2.2. Experimental design

*Acacia mangium* seedlings were produced from seeds of around 5-year old trees grown at the ICA/UFMG campus. Seeds were dried, dormancy-broken and treated with recommended bactericides/fungicides before sowing following standard protocol used for *Acacia* (= *Vachellia* Wight & Arn.) *farnesiana* (L.) Willd. in Brazil [35]. Seeds were sown in 8 × 12 cm plastic polybags (a seed per plastic polybag) in a nursery with its ruff covered using black shed net, with media mixing with 30% organic compost, 30% clay soil, 30% sand and 10% of reactive natural phosphate (160 g $seedling^{-1}$) in March 2014. The organic compost consisted of three parts, by volume: two parts of debris gardening pruning (≤ 5 cm) and one part of tanned nelore cattle *Bos taurus indicus* L., 1758 (Artiodactyla: Bovidae) manure. The mixture clay soil and sand was treated by a heating process at 80°C for 15 min. The soil pH of the pits was corrected with dolomitic limestone (i.e. an anhydrous carbonate mineral composed of calcium magnesium carbonate), increasing the base saturation to 50% [36]. Fritted trace elements (FTE), gypsum, micronutrients, natural phosphate and potassium chloride were added according to the soil chemical analysis for the Minas Gerais State [37]. Thirty-centimeter tall *A. mangium* seedlings were planted in pits (40 × 40 × 40 cm) spaced 2 m between them, in six parallel lines on flat terrain, spaced 2 m between lines, with four plants with and four without fertilization with dehydrated sewage sludge per line, in September 2014. These seedlings were irrigated twice a week until the beginning of the rainy season using water from a nearby river from when no additional water was provided. The plants were pruned using a razor sterilized with a solution of sodium hydroxide + sodium hypochlorite, when their branches reached 5 cm long, eliminating the additional shoots (i.e. others different from the leader shoot) and branches up to one-third of crown height, leaving only the leader shoot and lateral branches up to two-thirds of the crown height. The pruned parts of each plant (branches and shoots) were left between their respective planting lines. The experimental design was completely randomized in blocks with two treatments (20 l of dehydrated sewage sludge per pit or no dehydrated sewage sludge) and 24 replications with one plant each. The 20 l of dehydrated sewage sludge was placed in a single dose per pit at planting.

Dehydrated sewage sludge (5% moisture content) was collected at the sewage treatment plant—'Estação de Tratamento de Esgoto (ETE)' in the municipality of Juramento, Minas Gerais State, Brazil, about 40 km from the *A. mangium* experimental site. The ETE is operated by the Minas Gerais Sanitation Company S.A.—'Companhia de Saneamento de Minas Gerais S.A. (COPASA)' with capacity to treat 217 $m^3$ sewage sludge $d^{-1}$. The efficiency of the system in terms of removal of organic matter is higher than 90%. The sewage sludge goes through a solarization process in coarse sand tanks for three months in the ETE reducing the thermotolerant coliform bacteria to a level accepted by the National Council for the Environment—'Conselho Nacional do Meio Ambiente (CONAMA)' (Resolution N° 375) of the Ministry of the Environment—'Ministério do Meio Ambiente' of Brazil for use in agriculture, which is less than $10^3$ most likely number $g^{-1}$ of total solids. The main chemical and biological characteristics of the dehydrated sewage sludge of this company were pH–$H_2O$ = 4.40, N = 10.4 mg $Kg^{-1}$, P = 2.9 mg $Kg^{-1}$, K = 5.8 mg $Kg^{-1}$, Cd = 0.1 µg $g^{-1}$, Pb = 56.9 µg $g^{-1}$, Cr = 46.7 µg $g^{-1}$ and faecal coliforms = 4.35 most likely number $g^{-1}$ after analysis carried out in a laboratory [8].

## 2.3. Plant mass production and soil coverage

Leaves per branch, branches per plant, numbers and the percentage of soil cover by litter, grass and herbaceous plants were evaluated visually per month and plot ($1.0 \, m^2$), in the crown projection area of the 48 *A. mangium* plants.

## 2.4. Insects and spiders

Insects and spiders (no multiply counted) were counted by visual observation biweekly on the adaxial and abaxial surfaces of the leaves between 07.00 and 11.00 at the apical, middle and basal parts of the canopy in the northern, southern, eastern and western directions, totaling 12 leaves plant$^{-1}$ evaluation$^{-1}$ on the 48 *A. mangium* trees of six-month old for 24 months. Insects and spiders were not removed from plants during the evaluation, except those collected for identification. The total sample effort was 27 648 leaves covering the entire plant (vertical and horizontal axes), capturing as many insect and spider species as possible, especially the rarest. Insects and spiders present on the trunk (chest height) were collected, and insect defoliation was evaluated visually by the leaf area losses on a 0–100% scale with 5% increments for removed leaf area [38,39] for the 48 trees per evaluation. At least, three specimens per insect or spider species were captured per collection using aspirator, stored in glass flasks with 70% ethanol or mounted, separated into morphospecies and sent for identification.

## 2.5. Ecological indices

Averages were made by reducing the data to single trees. Ecological indices (diversity, individual abundance and species richness) were calculated for each identified species in the treatments (with or without dehydrated sewage sludge) per tree using the software BioDiversity Professional, Version 2 (©1997 The Natural History Museum) [40]. The diversity was calculated using Hill's formula [41,42] and the species richness with Simpson indices [43,44]. The predator (insects and spiders) and prey ratio on *A. mangium* was calculated per tree.

## 2.6. Statistics

Data on leaves per branch, branches per plant, percentages of soil cover by litter, grass and herbaceous plants, predator per prey ratio and defoliation, diversity, abundance and richness of chewing, defoliator and pollinator insect species, and arthropod predators (see the electronic supplementary material) were submitted to the non-parametric statistical hypothesis, Wilcoxon signed-rank test ($p < 0.05$) [45] using the statistical analysis program 'Sistema para Análises Estatísticas e Genéticas (SAEG)', version 9.1 [46] supplied by the 'Universidade Federal de Viçosa'.

The Spearman correlation matrix, among the most significant characteristics, was calculated. The matrices were submitted to correlation networks [47]. The edge thickness was controlled by applying a cut-of-value 0.26 (from which the Spearman correlation became significant, meaning that only edges with $|r_{ij}| \geq 0.26$ were highlighted). These analyses were performed using the R software version 3.4.1 by R Core Team [48]. The correlation network procedure was performed using the package qgraph [47].

# 3. Results

## 3.1. Leaves per branch, branches per plant, leaves per tree, percentages of defoliation and soil cover

Leaves per branch and branches per plant, percentages of defoliation by chewing insects and soil cover (litter, grasses and herbaceous plants) were higher for *A. mangium* plants fertilized with dehydrated sewage sludge than for those without fertilization, but no effect was observed on the predator per prey ratio (table 1). The increase in the number of leaves per tree reduced the predator per prey ratio (figure 1).

## 3.2. Biodiversity and richness indexes

The biodiversity and richness indexes for chewing and pollinator insects, and spiders only were similar for *A. mangium* plants fertilized or not with dehydrated sewage sludge. On the other hand, the abundance of chewing (greater than 10 times) and pollinator (greater than 2 times) insects, and total

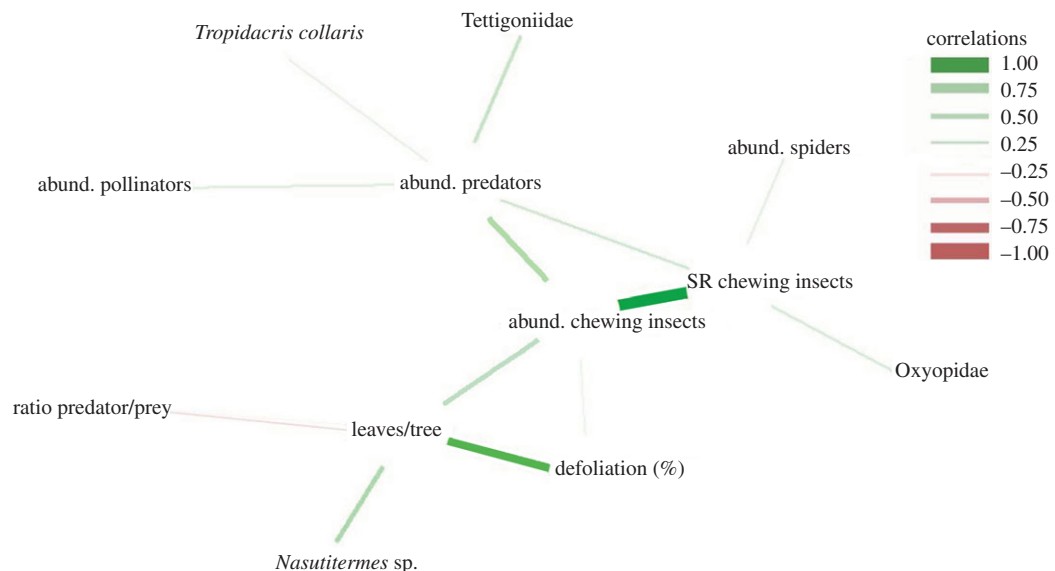

**Figure 1.** Estimated network structures based on Spearman correlation ($p < 0.05$) generated for abundance (abund.) of chewing, pollinator and predator insects, predator spiders, species richness (SR) of chewing insects, defoliation (%), ratio predator per prey, numbers of *Nasutitermes* sp. (Blattodea: Termitidae), Oxyopidae (Araneae), Tettigoniidae (Orthoptera) and *Tropidacris collaris* (Orthoptera: Romaleidae) on *Acacia mangium* (Fabales: Fabaceae) per tree $n = 48$.

**Table 1.** Numbers of total leaves per branch and branches per plant, defoliation, predator per prey ratio and soil cover (mean ± s.e.) of *Acacia mangium* (Fabales: Fabaceae) per tree with or without dehydrated sewage sludge. ($n = 24$ per treatment, VT = value of the test.)

|  | sewage sludge | | Wilcoxon test | |
| --- | --- | --- | --- | --- |
|  | without | with | VT | p |
| leaves per branch | 22.00 ± 0.69 | 33.71 ± 1.10 | 4.9 | 0.00 |
| branches per plant | 17.73 ± 0.56 | 41.26 ± 0.90 | 5.7 | 0.00 |
| defoliation (%) | 4.35 ± 0.25 | 6.28 ± 0.16 | 5.3 | 0.00 |
| predator per prey ratio | 10.37 ± 2.94 | 6.34 ± 1.49 | 1.4 | 0.08 |
| soil cover (%) | 8.43 ± 0.54 | 27.47 ± 1.10 | 5.6 | 0.00 |

predators (greater than 1.4 times) were higher on plants fertilized with dehydrated sewage sludge. Moreover, the treatments did not affect the spider ecological indices, but plants fertilized had more biodiversity of total predators (table 2). The increase in the number of leaves per tree increased the abundance and richness of chewing insects besides the predators on *A. mangium* plants. Higher species richness of chewing insects resulted in bigger predator abundances, including spiders (figure 1).

## 3.3. Arthropods

The termite *Nasutitermes* sp. (Blattodea: Termitidae) and the Neotropical stick grasshopper *Cephalocoema* sp. (Orthoptera: Proscopiidae) numbers were higher on *A. mangium* trunks fertilized with dehydrated sewage sludge and without fertilization, respectively, while the number of all other chewing insect species was similar ($p < 0.05$) between treatments. The Orthoptera chewers, the large South American grasshopper *Tropidacris collaris* Stoll, 1813 (Romaleidae) and the katydid Tettigoniidae and the Coleoptera *Lordops* sp. (Curculionidae) and *Stereoma anchoralis* Lacordaire, 1848 (Chrysomelidae) stood out in relation to the other chewing insects owing to their greater abundance on *A. mangium* plants, with or without dehydrated sewage sludge fertilization (table 3). The increase in the number of leaves per tree increased that of *Nasutitermes* sp. (figure 1).

The stingless bee, *Trigona spinipes* F., 1793 (Hymenoptera: Apidae) numbers were higher on *A. mangium* plants fertilized with dehydrated sewage sludge while those of the European honeybee, *Apis mellifera* L., 1758 and the stingless bee, *Tetragonisca angustula* Latreille, 1811 (Hymenoptera: Apidae) were similar ($p < 0.05$) between treatments, yet both had lower abundance than the first species (table 4).

**Table 2.** Diversity index (DI), species richness (SR) and abundance (abund.) of chewing and pollinator insects, total predators and spiders on *Acacia mangium* (Fabales: Fabaceae) per tree (mean ± SE) with or without dehydrated sewage sludge. (*n* = 24 per treatment, VT = value of the test.)

|  | sewage sludge | | Wilcoxon test | |
| --- | --- | --- | --- | --- |
|  | without | with | VT | *p* |
| DI chewing insects | 7.42 ± 1.35 | 4.66 ± 0.76 | 1.2 | 0.12 |
| SR chewing insects | 2.88 ± 0.40 | 3.17 ± 0.33 | 0.9 | 0.19 |
| abund. chewing insects | 3.96 ± 0.59 | 40.75 ± 17.80 | 2.3 | 0.01 |
| DI pollinators | 3.25 ± 0.37 | 2.74 ± 0.24 | 0.7 | 0.26 |
| SR pollinators | 1.25 ± 0.21 | 1.67 ± 0.14 | 0.2 | 0.43 |
| abund. pollinators | 2.96 ± 0.58 | 6.42 ± 1.28 | 2.2 | 0.02 |
| DI total predators | 14.96 ± 1.46 | 10.42 ± 1.32 | 2.3 | 0.01 |
| SR total predators | 7.88 ± 0.52 | 8.33 ± 0.52 | 0.1 | 0.45 |
| abund. total predators | 56.83 ± 17.59 | 77.92 ± 11.52 | 2.1 | 0.02 |
| DI spiders | 4.31 ± 0.38 | 3.59 ± 0.49 | 1.6 | 0.06 |
| SR spiders | 2.04 ± 0.19 | 1.75 ± 0.20 | 1.4 | 0.08 |
| abund. spiders | 2.50 ± 0.60 | 3.17 ± 0.39 | 1.0 | 0.17 |

The wasp *Polybia* sp. (Hymenoptera: Vespidae), the jumping spider Salticidae (Araneae) and the praying mantis *Mantis religiosa* (Linnaeus, 1758) (Mantodea: Mantidae) numbers were higher (*p* < 0.05) on *A. mangium* plants with or without, respectively, dehydrated sewage sludge, while those of all other predators insect and spider species was similar (*p* > 0.05) between treatments (table 4). The increase in the abundance of pollinators, Tettigoniidae and *T. collaris* individuals increased that of predators as well as the species richness of chewing insects resulted in higher numbers of the lynx spider Oxyopidae (Araneae) (figure 1).

## 4. Discussion

The highest abundance of chewing, pollinator and predator insects on *A. mangium* fertilized with dehydrated sewage sludge and soil covered by litter were owing to the better development of these plants (e.g. >leaves per tree), similar to that of the flooded gum *Eucalyptus grandis* W. Hill ex Maiden (Myrtales: Myrtaceae) [22] and their higher nitrogen levels in a dehydrated sewage sludge [8] obtained from the same ETE of the current study.

The litter cover in the crown projection area of *A. mangium* plants fertilized with dehydrated sewage sludge resulted from the higher number of leaves and branches produced by these plants compared to the non-fertilized ones, important for reducing laminar erosion and increasing soil fertility [49,50], confirming the first hypothesis in which fertilized plants are better in the recovery process of degraded areas. Dehydrated sewage sludge is rich in organic matter and macronutrients such as nitrogen and phosphorus, besides micronutrients such as copper and zinc, favouring tree growth and development [51,52]. The recovery of degraded areas is slow, but the use of *A. mangium* fertilized with dehydrated sewage sludge is promising, because of its fast growth and development, efficient fix atmospheric nitrogen gas fixation, potential to improve soil quality and widespread use [24]. This agrees with the positive impact of dehydrated sewage sludge in the development of the Brazilian pine, *Araucaria angustifolia* (Bertol) Kuntze (Pinales: Araucariaceae); Argentine cedar, *Cedrela fissilis* Vell. (Sapindales: Meliaceae); *E. grandis*; *Lafoensia pacari* St.-Hil. (Myrtales: Lythraceae), and *Senna spectabilis* (DC.) Irwin & Barneby (Fabales: Fabaceae) [22,23]. Dehydrated sewage sludge sanitized with neen, *Azadirachta indica* A. Juss. (Sapindales: Meliaceae) or with lime (a calcium-containing inorganic mineral) without this plant or in composition with *Ipomoea* sp. (Solanales: Convolvulaceae) compost with or without rock phosphate were evaluated. Other treatments included dehydrated sewage sludge with rock phosphate incorporated in the soil sanitized or not with *A. indica* or lime, and this fertilization with *Ipomoea* sp. compost with rock phosphate is incorporated in the soil. The density of pathogen (i.e. helminths and

**Table 3.** Total numbers of chewing insects per *Acacia mangium* (Fabales: Fabaceae) per tree (mean ± s.e.) with or without dehydrated sewage sludge. (*n* = 24 per treatment, VT = value of the test.)

| order: family | species | sewage sludge | | Wilcoxon test | |
|---|---|---|---|---|---|
| | | without | with | VT | p |
| Coleoptera: | | | | | |
| Buprestidae | *Psiloptera* sp. | 0.04 ± 0.04 | 0.00 ± 0.00 | 1.0 | 0.16 |
| Cerambycidae | non-identified | 0.08 ± 0.05 | 0.00 ± 0.00 | 1.4 | 0.08 |
| Chrysomelidae | *Alagoasa* sp. | 0.04 ± 0.04 | 0.04 ± 0.04 | 0.0 | 0.50 |
| | *Cerotoma* sp. | 0.17 ± 0.07 | 0.21 ± 0.08 | 0.4 | 0.36 |
| | *Diabrotica speciosa* Germar | 0.08 ± 0.05 | 0.25 ± 0.09 | 1.5 | 0.06 |
| | *Disonycha brasiliensis* Lima | 0.04 ± 0.04 | 0.04 ± 0.04 | 0.0 | 0.50 |
| | *Eumolpus* sp. | 0.04 ± 0.04 | 0.13 ± 0.06 | 1.0 | 0.15 |
| | *Lamprosoma* sp. | 0.04 ± 0.04 | 0.00 ± 0.00 | 1.0 | 0.16 |
| | *Parasyphraea* sp. | 0.25 ± 0.12 | 0.46 ± 0.19 | 1.0 | 0.17 |
| | *Stereoma anchoralis* Lacordaire | 0.46 ± 0.21 | 0.21 ± 0.13 | 0.8 | 0.20 |
| | *Walterianella* sp. | 0.04 ± 0.04 | 0.04 ± 0.04 | 0.0 | 0.50 |
| | *Wanderbiltiana* sp. | 0.04 ± 0.04 | 0.04 ± 0.04 | 0.0 | 0.50 |
| Curculionidae | *Lordops* sp. | 0.00 ± 0.00 | 0.46 ± 0.41 | 1.4 | 0.08 |
| | non-identified | 0.04 ± 0.04 | 0.08 ± 0.05 | 0.6 | 0.28 |
| Tenebrionidae | Alleculinae | 0.04 ± 0.04 | 0.00 ± 0.00 | 1.0 | 0.16 |
| | *Epitragus* sp. | 0.04 ± 0.04 | 0.04 ± 0.04 | 0.0 | 0.50 |
| Blattodea: | | | | | |
| Termitidae | *Nasutitermes* sp.[a] | 0.00 ± 0.00 | 36.37 ± 14.73 | 3.3 | 0.00 |
| Lepidoptera: | | | | | |
| non-identified | non-identified | 0.33 ± 0.13 | 0.21 ± 0.10 | 0.7 | 0.23 |
| Orthoptera: | | | | | |
| Gryllidae | non-identified | 0.04 ± 0.04 | 0.04 ± 0.04 | 0.0 | 0.50 |
| Tettigoniidae | non-identified | 1.00 ± 0.19 | 0.75 ± 0.18 | 1.0 | 0.16 |
| Proscopiidae | *Cephalocoema* sp. | 0.13 ± 0.06 | 0.00 ± 0.00 | 1.8 | 0.04 |
| Romaleidae | *Tropidacris collaris* Stoll | 1.00 ± 0.19 | 1.33 ± 0.26 | 0.7 | 0.25 |
| Phasmatodea: | | | | | |
| Phasmidae | *Phibalosoma phyllinum* Gray | 0.00 ± 0.00 | 0.04 ± 0.04 | 1.0 | 0.16 |

[a]Observed on *A. mangium* trunk.

protozoans) in these treatments was low and similar with soils without fertilization or with liming and chemical fertilization [8]. These treatments did not surpass the maximum limits of annual addition and the permissible maximum levels of heavy metal concentrations in the soils, but the concentrations of lead in *Z. mays* and *V. unguiculata* grains reached values above the limits permitted for agricultural products, regardless of the addition of sewage sludge in the soil [8].

The greater abundance of chewing insects and defoliation on *A. mangium* plants fertilized with dehydrated sewage sludge is probably owing to the greater number of leaves serving as a better food source and quality for insects. This confirms the second hypothesis that the diversity and abundance of herbivorous insects and their predators are usually higher and with higher increase of chewing insects than predators (e.g. >leaves ≤predator per prey ratio) on trees with higher leaf mass [25,53,54]. These trees function as a BGI, but with a higher chance of rare species extinction on those with lower leaf mass [25,28,55]. In addition, the quantity of free amino acids and proteins is superior in plants with higher nitrogen fertilization, favouring herbivorous insects [56]. Interactions between insects and *Acacia* species plants show the potential of this plant to increase the biodiversity and recover

**Table 4.** Total numbers of spiders, and insect predators and pollinators per *Acacia mangium* (Fabales: Fabaceae) per tree (mean ± s.e.) with or without dehydrated sewage sludge. (*n* = 24 per treatment, VT = value of the test.)

| order: family | species | sewage sludge | | Wilcoxon test | |
|---|---|---|---|---|---|
| | | without | with | VT | p |
| **Araneae:** | | | | | |
| Anyphaenidae | *Teudis* sp. | 0.04 ± 0.04 | 0.00 ± 0.00 | 1.0 | 0.16 |
| Araneidae | non-identified | 1.21 ± 0.22 | 0.83 ± 0.19 | 1.3 | 0.10 |
| Oxyopidae | non-identified | 0.50 ± 0.17 | 0.75 ± 0.19 | 1.2 | 0.11 |
| | *Oxyopes salticus* Hentz | 0.17 ± 0.07 | 0.08 ± 0.05 | 0.9 | 0.19 |
| Salticidae | non-identified | 1.08 ± 0.33 | 0.54 ± 0.19 | 1.8 | 0.04 |
| | *Uspachus* sp. | 0.04 ± 0.04 | 0.08 ± 0.08 | 0.0 | 0.49 |
| Sparassidae | *Quemedice* sp. | 0.04 ± 0.04 | 0.04 ± 0.04 | 0.0 | 0.50 |
| Tetragnathidae | *Leucauge* sp. | 0.04 ± 0.04 | 0.08 ± 0.08 | 1.0 | 0.16 |
| Thomisidae | *Aphantochilus rogersi* Cambridge | 0.04 ± 0.04 | 0.04 ± 0.04 | 0.0 | 0.50 |
| | *Tmarus* sp. | 0.04 ± 0.04 | 0.04 ± 0.04 | 0.0 | 0.50 |
| **Hemiptera:** | | | | | |
| Pentatomidae | *Podisus* sp. | 0.04 ± 0.04 | 0.13 ± 0.06 | 1.0 | 0.15 |
| **Hymenoptera:** | | | | | |
| Apidae | *Apis mellifera* L. | 0.50 ± 0.19 | 0.38 ± 0.15 | 0.2 | 0.44 |
| | *Tetragonisca angustula* Latreille | 1.17 ± 0.26 | 1.38 ± 0.24 | 0.8 | 0.22 |
| | *Trigona spinipes* F. | 1.29 ± 0.39 | 4.67 ± 1.31 | 2.2 | 0.02 |
| Vespidae | *Polybia* sp. | 0.60 ± 0.26 | 3.75 ± 2.71 | 2.1 | 0.02 |
| **Mantodea:** | | | | | |
| Mantidae | *Mantis religiosa* L. | 0.25 ± 0.09 | 0.04 ± 0.04 | 2.0 | 0.02 |

degraded areas around the world [57–60]. The dehydrated sewage sludge as a biofertilizer improved macrofauna recovery, including the scarab beetles Scarabaeidae (Coleoptera) larvae and adults in degraded soils of the Cerrado (Brazilian savannah) type biome area [55].

The presence of *Nasutitermes* sp., as the most abundant insect on *A. mangium* plant trunks fertilized with dehydrated sewage sludge may be owing to the organic matter richness of this fertilizer [8,51] and the higher litter production by this plant (e.g. >leaves ≥ *Nasutitermes* sp.). This insect can damage living or dead trees and processed wood, including root systems, although they caused galleries in the trunks without damaging or causing plant death [61]. Damage by *Lordops* sp., *S. anchoralis*, *T. collaris* and Tettigoniidae on *A. mangium* leaves and their greater abundance compared to that of other chewing insects is worrying. *Tropidacris collaris* damaged the swamp she-oak, *Casuarina glauca* Sieber (Fagales: Casuarinaceae) and white leadtree, *Leucaena leucocephala* (Lam.) de Wit (Fabales: Fabaceae) [62,63]. *Meroncidius intermedius* Brunner Von Wattenwyl, 1895 (Orthoptera: Tettigonniidae) damaged grasses and banana *Musa* spp. fruits (Zingiberales: Musaceae) [64] and *Lordops* sp. defoliated the diesel tree, *Copaifera langsdorffii* Desf. (Fabales: Fabaceae) [65], but there is no reports of *S. anchoralis* damaging commercial plants.

The number of pollinating insects being two times higher on *A. mangium* plants fertilized with dehydrated sewage sludge is probably owing to their larger canopy size, higher number of flowers and supporting a greater insect numbers [25,54], including pollinators, confirming the third hypothesis: greater BGI greater pollinating insects. In addition, nitrogen fertilization via dehydrated sewage sludge may have increased the pollen and/or nectar production and quality (more amino acids and protein) in *A. mangium* flowers, increasing pollinator attractiveness as observed for the higher attraction of *Nicotiana* L. (Solanales: Solanaceae) species flowers with better quality of nectar sugars and amino acids by different groups of pollinators (e.g. bats (Chiroptera), hummingbirds (Trochilidae) or moths (Lepidoptera)) in Wuppertal, Germany [66] and floral pollens with better quality by bumblebees, *Bombus* Latreille, 1802 (Hymenoptera: Apidae) and honeybees, *Aphis* species in Newcastle, United Kingdon [67]. The greater *T. spinipes* pollinator abundance on *A. mangium* plants, especially on those

fertilized with dehydrated sewage sludge may be of low importance, because it can reduce pollination as reported for Cucurbitaceae (Cucurbitales) owing to insufficient pollen transportation (small body size) and/or chasing other pollinators, such as *A. mellifera* and *T. angustula*, by flying in flocks and with aggressive behaviour [68]. In addition, *T. spinipes* damages shoots and plant growth regions by removing fibres to construct their nests, as reported on *A. mangium* and *L. leucocephala*, that also had their leaves and shoots damaged [63,69].

The greater abundance of predator insects and spiders, on *A. mangium* plants fertilized with dehydrated sewage sludge, is probably owing to the higher number of chewing and pollinator insects on the plants (larger trees), that is, these predators followed their prey [70], confirming the fourth hypothesis: greater BGI greater predators. In general, the number of predator insect and spider species did not differ between *A. mangium* plants with or without dehydrated sewage sludge fertilization, but the number of spider species (30% higher on fertilized plants) and the abundance of the predatory wasp *Polybia* sp. were higher on fertilized plants. Spider predators reduced insect damage, mainly from defoliators, such as spiders in many agroecosystems in the USA [71], wolf (Araneae: Lycosidae) and sheet weaver (Araneae: Linyphiidae) spiders in winter barley, *Hordeum vulgare* L. (Poales: Poaceae) fields situated in differently structured landscapes in Uppsala, Sweden [72], wandering spiders (Araneae: Ctenidae) in agroecosystems in Italy [73] and spiders in pequi, *Caryocar brasiliense* Cambess. (Malpighiales: Caryocaraceae) trees in Minas Gerais State, Brazil [26]. Predatory wasps (Vespidae) are important natural enemies in agricultural systems such as *Brassica campestris* L. and kale, *Brassica oleracea* L. var. *acephala* DC., Arabian coffee, *Coffea arabica* L. (Gentianales: Rubiaceae) and tomato, *Solanum lycopersicon* L. (Solanales: Solanaceae), preying mainly on caterpillars and leaf miners (Lepidoptera) in several regions of Brazil [74–77]. Sewage sludge increased the ground beetle Carabidae (Coleoptera) species richness in the area of Oxford, USA [78].

# 5. Conclusion

To summarize, the larger *A. mangium* crown (>BGI) fertilized with dehydrated sewage sludge increases soil cover (e.g. litter) and the abundance of chewing (>defoliation) and pollinator insects and arthropod predators, showing that this plant is adequate for recovering degraded areas using this fertilization. The presence of *Nasutitermes* sp. on *A. mangium* plant trunks fertilized with dehydrated sewage sludge may be owing to the organic matter richness of this fertilizer and the higher litter production by this plant (e.g. >leaves ≥ *Nasutitermes* sp.), but without damaging or causing plant death. On the other hand, *Lordops* sp., *T. collaris* and Tettigoniidae damaged leaves of *A. mangium* and this is worrying because these insects are pests in other economically important crops. The greater *T. spinipes* pollinator abundance on *A. mangium* plants is a problem owing to this insect damaged their shoots and plant growth regions.

Ethics. No specific permits are required to plant *Acacia mangium* in Brazil. The laboratory and field studies did not involve endangered or protected species.

Data accessibility. All data generated or analysed during this study are included in this manuscript.

Authors' contributions. The study was conceived and designed by J.L.S., G.L.D.L. and R.A.S., data were collected by J.L.S., W. de S.T. and F.W.S.S. with support from G.L.D.L. and R.A.S., data analysis was done by G.L.D.L. and A.M.A., figures, tables and manuscript preparation was done by J.L.S., G.L.D.L., W. de S.T., A.M.A., J.E.S. and J.C.Z. and all authors contributed to revisions and approved the final manuscript.

Competing interests. The authors declare that they have no conflict of interest.

Funding. The study was financially supported by the following Brazilian agencies 'Conselho Nacional de Desenvolvimento Científico e Tecnológico (CNPq)', 'Coordenação de Aperfeiçoamento de Pessoal de Nível Superior (CAPES-Finance Code 001)', 'Fundação de Amparo à Pesquisa do Estado de Minas Gerais (FAPEMIG)' and 'Programa Cooperativo sobre Proteção Florestal (PROTEF)' of the 'Instituto de Pesquisas e Estudos Florestais (IPEF)'.

Acknowledgements. We would like to thank Dr Antônio Domingos Brescovit (Instituto Butantan, São Paulo State, Brazil—Arachnida) and Dr Ayr de Moura Bello (Fundação Oswaldo Cruz, Rio de Janeiro State, Brazil—Coleoptera) for arthropod species identification. The voucher numbers for insects are 1595/02 and 1597/02 (Centro de Estudos Faunísticos e Ambientais, Universidade Federal do Paraná, Curitiba, Paraná State, Brazil) and that of spiders is IBSP 36921–36924 (Instituto Butantan, São Paulo State, Brazil). Dr Phillip John Villani (University of Melbourne, Australia) revised and corrected the English language used in this manuscript.

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
