## [Reviewer comments · Royal Society Open Science]

Review History

RSOS-191196.R0 (Original submission)

Review form: Reviewer 1

Is the manuscript scientifically sound in its present form?

Yes

Are the interpretations and conclusions justified by the results?

Yes

Is the language acceptable?

Yes

Do you have any ethical concerns with this paper?

No

Have you any concerns about statistical analyses in this paper?

No

Recommendation?

Major revision is needed (please make suggestions in comments)

Comments to the Author(s)

The authors explore the role of dehydrated sewage fertilization in driving abundance and diversity of arthropods, in addition to plant growth. Arthropod groups were visually identified, with voucher specimens collected, over 24 months on 6 month old *Acacia magnium* trees. The trees were found to grow more leaves and branches per plant in the sewage treatment. Further, abundance of several chewing, pollinator, and predator arthropod groups were greater in the sewage treatment.

This study examines a potentially impactful question, relevant for a range of ecological practitioners. The use of by-product sewage sludge in ecosystem restoration seems to be a growing solution of interest. My major concerns pertain to the establishment treatment block, limits of inference from the arthropod surveying protocol applied, and use of only two treatment levels (control and supplementation).

Pertaining to the first concern, this study appears to have fairly high number of replicates in the two levels of treatment, but the replicates were spaced two meters apart. It could be questioned if this distance was sufficient to prevent seepage from the supplementation and/or interactions between adjacent plants.

The arthropod surveys appeared to be consistently conducted at 7:00 and 11:00 am on six month old trees, for the duration of 24 months. This seems to lend fairly strong support for the inferences that were documented in the study results. However, this protocol and documented findings seem strongly biased towards diurnal arthropod groups. This potentially led to underestimation of ecologically important nocturnal groups, such as many other arachnids, as activity might have not been detected by visual surveys when those groups were in refuges. Further, from the manuscript it is difficult to determine if flowers and/or fruiting bodies developed in the *Acacia* during the study, which may have attracted additional arthropod diversity.

The study also seems to have been conducted on single population with only two treatment levels, supplementation or non-supplementation at a single point in time. The authors did seem to have robustly characterized the sewage being used for supplementation (e.g. Lines 138-141) and this is at least one open area for research. It is difficult to assess whether supplementation at intermediate levels of sewage would increase or diminish the observed responses in plants and arthropods. Another area ripe for future study will be measures of how sewage supplementation drives nutrition of plants, including parsing nitrogen compound transfers from the sewage to plant amino acids and proteins (e.g. leaf and fruit content) compared to other forms of fertilizers, and measures partitioning which sewage nutrients are associated with nutrient uptake/cycling by different arthropod consumers groups.

Specific Comments:

1. Line 70-79: Invoking the Biogeographic Island Theory seems intriguing, but perhaps difficult to conceptually support or refute in this study. The optimal foraging framework, such as relating resource "patch" sizes and arthropod detection (e.g. crowns, litter bunches, or other plant nutrient hotspots), could more directly relate to the ecological scale of arthropod foraging and contributions of sewage to nutrient cycling.
2. Lines 144-146: Were plant flowers or fruiting bodies counted?
3. Lines 149-152: Does "no multiply counted" refer to not counting an arthropod multiple times within any given 7:00/10:00am survey or does it mean that arthropods were marked to prevent later re-counting?
4. Lines 170-181: What was the justification for using the non-parametric alternative, Wilcoxon signed-rank test, for the two-treatment comparisons?

Review form: Reviewer 2

Is the manuscript scientifically sound in its present form?

Yes

Are the interpretations and conclusions justified by the results?

Yes

Is the language acceptable?

Yes

Do you have any ethical concerns with this paper?

No

Have you any concerns about statistical analyses in this paper?

No

Recommendation?

Major revision is needed (please make suggestions in comments)

Comments to the Author(s)

The manuscript is well prepared and very well written. Materials and methods are consistent with the proposed objectives as well as the hypothesis tested. The results are clear and very important to the research line. Discussion is objective and coherent. References are current and consistent with the theme. Finally, I congratulate the authors for their valuable contribution to the journal.

Decision letter (RSOS-191196.R0)

09-Oct-2019

Dear Dr Tavares,

The editors assigned to your paper ("Diversity of arthropods on *Acacia mangium* (Fabaceae) and production of this plant with dehydrated sewage sludge in degraded area") have now received comments from reviewers. We would like you to revise your paper in accordance with the referee and Associate Editor suggestions which can be found below (not including confidential reports to the Editor). Please note this decision does not guarantee eventual acceptance.

Please submit a copy of your revised paper before 01-Nov-2019. Please note that the revision deadline will expire at 00.00am on this date. If we do not hear from you within this time then it will be assumed that the paper has been withdrawn. In exceptional circumstances, extensions may be possible if agreed with the Editorial Office in advance. We do not allow multiple rounds of revision so we urge you to make every effort to fully address all of the comments at this stage. If deemed necessary by the Editors, your manuscript will be sent back to one or more of the original reviewers for assessment. If the original reviewers are not available, we may invite new reviewers.

To revise your manuscript, log into <http://mc.manuscriptcentral.com/rsos> and enter your Author Centre, where you will find your manuscript title listed under "Manuscripts with

Decisions." Under "Actions," click on "Create a Revision." Your manuscript number has been appended to denote a revision. Revise your manuscript and upload a new version through your Author Centre.

- Data accessibility

If you wish to submit your supporting data or code to Dryad (<http://datadryad.org/>), or modify your current submission to dryad, please use the following link:
<http://datadryad.org/submit?journalID=RSOS&manu=RSOS-191196>

- Competing interests

- Authors' contributions

- Acknowledgements

- Funding statement

Kind regards,

Lianne Parkhouse
Royal Society Open Science
openscience@royalsociety.org

on behalf of Dr Punidan Jeyasingh (Associate Editor) and Professor Kevin Padian (Subject Editor)
openscience@royalsociety.org

Associate Editor's comments (Dr Punidan Jeyasingh):

This manuscript reports data on the abundance and diversity of arthropods on Acacia that were treated with sewage sludge. This is an important aspect of research. The manuscript was reviewed by two reviewers, both of whom were positive about the study question and its implications. Nevertheless, one reviewer raised a few legitimate issues that the authors needs to address in a revision. I felt the reviews were fair and constructive. I invite the authors to address these points in a revision.

Reviewers' Comments to Author:

Reviewer: 1

Comments to the Author(s)

The authors explore the role of dehydrated sewage fertilization in driving abundance and diversity of arthropods, in addition to plant growth. Arthropod groups were visually identified, with voucher specimens collected, over 24 months on 6 month old Acacia magnium trees. The trees were found to grow more leaves and branches per plant in the sewage treatment. Further, abundance of several chewing, pollinator, and predator arthropod groups were greater in the sewage treatment.

This study examines a potentially impactful question, relevant for a range of ecological practitioners. The use of by-product sewage sludge in ecosystem restoration seems to be a growing solution of interest. My major concerns pertain to the establishment treatment block, limits of inference from the arthropod surveying protocol applied, and use of only two treatment levels (control and supplementation).

Pertaining to the first concern, this study appears to have fairly high number of replicates in the two levels of treatment, but the replicates were spaced two meters apart. It could be questioned if this distance was sufficient to prevent seepage from the supplementation and/or interactions between adjacent plants.

The arthropod surveys appeared to be consistently conducted at 7:00 and 11:00 am on six month old trees, for the duration of 24 months. This seems to lend fairly strong support for the

inferences that were documented in the study results. However, this protocol and documented findings seem strongly biased towards diurnal arthropod groups. This potentially led to underestimation of ecologically important nocturnal groups, such as many other arachnids, as activity might have not been detected by visual surveys when those groups were in refuges. Further, from the manuscript it is difficult to determine if flowers and/or fruiting bodies developed in the Acacia during the study, which may have attracted additional arthropod diversity.

The study also seems to have been conducted on single population with only two treatment levels, supplementation or non-supplementation at a single point in time. The authors did seem to have robustly characterized the sewage being used for supplementation (e.g. Lines 138-141) and this is at least one open area for research. It is difficult to assess whether supplementation at intermediate levels of sewage would increase or diminish the observed responses in plants and arthropods. Another area ripe for future study will be measures of how sewage supplementation drives nutrition of plants, including parsing nitrogen compound transfers from the sewage to plant amino acids and proteins (e.g. leaf and fruit content) compared to other forms of fertilizers, and measures partitioning which sewage nutrients are associated with nutrient uptake/cycling by different arthropod consumers groups.

Specific Comments:

1. Line 70-79: Invoking the Biogeographic Island Theory seems intriguing, but perhaps difficult to conceptually support or refute in this study. The optimal foraging framework, such as relating resource "patch" sizes and arthropod detection (e.g. crowns, litter bunches, or other plant nutrient hotspots), could more directly relate to the ecological scale of arthropod foraging and contributions of sewage to nutrient cycling.
2. Lines 144-146: Were plant flowers or fruiting bodies counted?
3. Lines 149-152: Does "no multiply counted" refer to not counting an arthropod multiple times within any given 7:00/10:00am survey or does it mean that arthropods were marked to prevent later re-counting?
4. Lines 170-181: What was the justification for using the non-parametric alternative, Wilcoxon signed-rank test, for the two-treatment comparisons?

Reviewer: 2

Comments to the Author(s)

The manuscript is well prepared and very well written. Materials and methods are consistent with the proposed objectives as well as the hypothesis tested. The results are clear and very important to the research line. Discussion is objective and coherent. References are current and consistent with the theme. Finally, I congratulate the authors for their valuable contribution to the journal.

Author's Response to Decision Letter for (RSOS-191196.R0)

See Appendix A.

RSOS-191196.R1 (Revision)

Review form: Reviewer 1

Is the manuscript scientifically sound in its present form?

Yes

Are the interpretations and conclusions justified by the results?

Yes

Is the language acceptable?

Yes

Do you have any ethical concerns with this paper?

No

Have you any concerns about statistical analyses in this paper?

No

Recommendation?

Accept as is

Comments to the Author(s)

The authors seemed to have provided adequate clarification to my previous review comments. That is, the authors provide justification for their treatment interspersed distance, sampling protocol (diurnal, citing safety concerns), why additional response variables weren't measured (e.g. no fruiting bodies present), and also for their use of non-parametric statistical procedure. Further, the authors provided forethought into how this research could lead to the design of future studies (e.g. dosage dependence of sewage and potential nutritive effects). As mentioned in the previous round of review, I strongly consider the research to be interesting and potentially impactful. The thoughtful rebuttal and manuscript revisions in-text seem to further support this.

Decision letter (RSOS-191196.R1)

03-Feb-2020

Dear Dr Zanuncio,

It is a pleasure to accept your manuscript entitled "Diversity of arthropods on *Acacia mangium* (Fabaceae) and production of this plant with dehydrated sewage sludge in degraded area" in its current form for publication in Royal Society Open Science. The comments of the reviewer(s) who reviewed your manuscript are included at the foot of this letter.

You can expect to receive a proof of your article in the near future. Please contact the editorial

office (openscience_proofs@royalsociety.org) and the production office (openscience@royalsociety.org) to let us know if you are likely to be away from e-mail contact -- if you are going to be away, please nominate a co-author (if available) to manage the proofing process, and ensure they are copied into your email to the journal.

on behalf of Dr Punidan Jeyasingh (Associate Editor) and Kevin Padian (Subject Editor)
openscience@royalsociety.org

Associate Editor Comments to Author (Dr Punidan Jeyasingh):

Comments to the Author:

I thank the authors for addressing all reviewer comments. This version is much improved compared to the original one submitted. With much gratitude to the expert reviewers, I am happy to recommend publication of this manuscript.

Reviewer comments to Author:

Reviewer: 1

Comments to the Author(s)

The authors seemed to have provided adequate clarification to my previous review comments. That is, the authors provide justification for their treatment interspersed distance, sampling protocol (diurnal, citing safety concerns), why additional response variables weren't measured (e.g. no fruiting bodies present), and also for their use of non-parametric statistical procedure. Further, the authors provided forethought into how this research could lead to the design of future studies (e.g. dosage dependence of sewage and potential nutritive effects). As mentioned in the previous round of review, I strongly consider the research to be interesting and potentially impactful. The thoughtful rebuttal and manuscript revisions in-text seem to further support this.

Appendix A

1 Corrections Letter

2

3 Dear Editor-in-Chief, Professor Dr. Jeremy Sanders

4 CEB FRS

5

6 Please find attached the revised version of our manuscript entitled “Diversity of
7 arthropods on *Acacia mangium* (Fabaceae) and production of this plant with dehydrated
8 sewage sludge in degraded area”. This version consists of 7703 words, including
9 references, 1 figure and 4 tables.

10 We are really pleased with the positive comments our manuscript has attracted
11 from all reviewers. It is worth mention that all comments were addressed and helped us
12 to improve the manuscript quality, particularly establishment treatment block, limits of
13 inference from the arthropod surveying protocol applied and use of only two treatment
14 levels (control and supplementation).

15 We have addressed point-by-point the issues rose by the reviewers (original
16 reviewer comments in regular typeface, responses in boldface). We believe that our
17 manuscript has achieved the Royal Society Open Science quality standards. We look
18 forward to receive your decision and we are ready to make any other corrections that
19 you consider necessary.

20 Please note that we have included Professor José Eduardo Serrão as a co-author
21 of the manuscript because he has contributed with corrections on the manuscript
22 suggested by the Reviewers. All authors agreed to include Professor Serrão as a co-
23 author.

24

25 Yours sincerely,

26

27 Wagner de Souza Tavares

28

29 Reviewers' Comments to Author:

30 Reviewer: 1

31 Comments to the Author(s)

32

33 The authors explore the role of dehydrated sewage fertilization in driving abundance
34 and diversity of arthropods, in addition to plant growth. Arthropod groups were visually

35 identified, with voucher specimens collected, over 24 months on 6 month old *Acacia*
36 *magnium* trees. The trees were found to grow more leaves and branches per plant in the
37 sewage treatment. Further, abundance of several chewing, pollinator, and predator
38 arthropod groups were greater in the sewage treatment.

39

40 This study examines a potentially impactful question, relevant for a range of ecological
41 practitioners. The use of by-product sewage sludge in ecosystem restoration seems to be
42 a growing solution of interest. My major concerns pertain to the establishment treatment
43 block, limits of inference from the arthropod surveying protocol applied, and use of
44 only two treatment levels (control and supplementation).

45

46 Pertaining to the first concern, this study appears to have fairly high number of
47 replicates in the two levels of treatment, but the replicates were spaced two meters apart.
48 It could be questioned if this distance was sufficient to prevent seepage from the
49 supplementation and/or interactions between adjacent plants.

50 **Answer: The distance was sufficient, especially when the difference on the**
51 **parameters of leaves, branches and litter produced between fertilized and non-**
52 **fertilized plants is clearly seen.**

53 **The area is quite degraded and the climate is characteristic of hot and dry**
54 **region - semi arid; therefore, the tighter spacing utilized between plants aimed to**
55 **reduce desiccation caused by the hot and dry winds coming from the northeastern**
56 **semi-arid in our region.**

57

58 The arthropod surveys appeared to be consistently conducted at 7:00 and 11:00 am on
59 six month old trees, for the duration of 24 months. This seems to lend fairly strong
60 support for the inferences that were documented in the study results. However, this
61 protocol and documented findings seem strongly biased towards diurnal arthropod
62 groups. This potentially led to underestimation of ecologically important nocturnal
63 groups, such as many other arachnids, as activity might have not been detected by
64 visual surveys when those groups were in refuges. Further, from the manuscript it is
65 difficult to determine if flowers and/or fruiting bodies developed in the *Acacia* during
66 the study, which may have attracted additional arthropod diversity.

67 **Answer: The study aimed to evaluate the diurnal insects. The nocturnal insects**
68 **would be very difficult to evaluate. The use of light at night may attract non-**

69 **associated insects to *A. mangium*; moreover, our region is rich in venomous pit**
70 **vipers and coral snakes of nocturnal habits such as jararaca, *Bothrops jararaca***
71 **(Wied-Neuwied) (Squamata: Viperidae), cascavel, *Crotalus durissus* Linnaeus**
72 **(Squamata: Viperidae) and *Micrurus brasiliensis* (Roze) (Squamata: Elapidae),**
73 **which could put our team at risk of life.**

74 **There were no flowers and, consequently, fruits on the plants as they were**
75 **still relatively young and in a semi-arid climate - which reduces the annual growth**
76 **rate and the entry into the reproductive period - allied to the degraded soil in the**
77 **area.**

78
79 The study also seems to have been conducted on single population with only two
80 treatment levels, supplementation or non-supplementation at a single point in time. The
81 authors did seem to have robustly characterized the sewage being used for
82 supplementation (e.g. Lines 138-141) and this is at least one open area for research. It is
83 difficult to assess whether supplementation at intermediate levels of sewage would
84 increase or diminish the observed responses in plants and arthropods. Another area ripe
85 for future study will be measures of how sewage supplementation drives nutrition of
86 plants, including parsing nitrogen compound transfers from the sewage to plant amino
87 acids and proteins (e.g. leaf and fruit content) compared to other forms of fertilizers, and
88 measures partitioning which sewage nutrients are associated with nutrient
89 uptake/cycling by different arthropod consumers groups.

90 **Answer: We agree with the above weights. This experiment aimed to test, as the**
91 **first stage, one dosage considered high and the other with nothing - control. Future**
92 **experiment will aim, since the sludge had positive effect on the plants, intermediate**
93 **dosages. Yes, it may, in further works, studying the effect of sewage sludge on free**
94 **amino acids and proteins, such as insect colonization.**

95 **We emphasize that this experiment was the first, the basis for future work,**
96 **in which we wanted to test if there would be effect of sludge on the plant - more**
97 **leaves - helping faster the recovery of degraded soil in the area - as well as the**
98 **effect on insects. We believe that the present work was quite satisfactory in its**
99 **proposed objectives.**

100

101 Specific Comments:

102

103 1. Line 70-79: Invoking the Biogeographic Island Theory seems intriguing, but perhaps
104 difficult to conceptually support or refute in this study. The optimal foraging
105 framework, such as relating resource “patch” sizes and arthropod detection (e.g. crowns,
106 litter bunches, or other plant nutrient hotspots), could more directly relate to the
107 ecological scale of arthropod foraging and contributions of sewage to nutrient cycling.

108 **Answer: We consider a tree as a biogeographic island for insects, which is possible**
109 **in the literature, because for many insects, especially the smallest and/or low**
110 **dispersion, that plant, for its, works as an island.**

111

112 2. Lines 144-146: Were plant flowers or fruiting bodies counted?

113 **Answer: No flowers and fruiting bodies were evaluated.**

114

115 3. Lines 149-152: Does “no multiply counted” refer to not counting an arthropod
116 multiple times within any given 7:00/10:00am survey or does it mean that arthropods
117 were marked to prevent later re-counting?

118 **Answer: This means that the insect was not counted twice or more in that**
119 **assessment. Insects were not marked.**

120

121 4. Lines 170-181: What was the justification for using the non-parametric alternative,
122 Wilcoxon signed-rank test, for the two-treatment comparisons?

123 **Answer: It used the nonparametric test because the data do not present normal**
124 **distribution, a fact that is normal for insects, especially under field conditions,**
125 **where many zero appear and in some samples large numbers of insects (gregarious**
126 **attack).**

127

128 Reviewer: 2

129 Comments to the Author(s)

130

131 The manuscript is well prepared and very well written. Materials and methods are
132 consistent with the proposed objectives as well as the hypothesis tested. The results are
133 clear and very important to the research line. Discussion is objective and coherent.
134 References are current and consistent with the theme. Finally, I congratulate the authors
135 for their valuable contribution to the journal.